# Effects of tDCS on the attentional blink revisited: A statistical evaluation of a replication attempt

**Leon C. Reteig**[1,2], **Lionel A. Newman**[1,3], **K. Richard Ridderinkhof**[1,2], **Heleen A. Slagter**[1,4] *

**1** Department of Psychology, University of Amsterdam, Amsterdam, The Netherlands, **2** Amsterdam Brain and Cognition, University of Amsterdam, Amsterdam, The Netherlands, **3** Department of Artificial Intelligence and Cognitive Engineering, University of Groningen, Groningen, The Netherlands, **4** Department of Applied and Experimental Psychology, Vrije Universiteit Amsterdam, Amsterdam, The Netherlands

* h.a.slagter@vu.nl

**Data Availability Statement:** All of the data and materials from this study and the data from London and Slagter (2021) are available on the Open Science Framework: https://doi.org/10. 17605/OSF.IO/Y6HSF The analysis code is

## Abstract

The attentional blink (AB) phenomenon reveals a bottleneck of human information processing: the second of two targets is often missed when they are presented in rapid succession among distractors. In our previous work, we showed that the size of the AB can be changed by applying transcranial direct current stimulation (tDCS) over the left dorsolateral prefrontal cortex (lDLPFC) (London & Slagter, *Journal of Cognitive Neuroscience*, *33*, 756–68, 2021). Although AB size at the group level remained unchanged, the effects of anodal and cathodal tDCS were negatively correlated: if a given individual's AB size decreased from baseline during anodal tDCS, their AB size would increase during cathodal tDCS, and vice versa. Here, we attempted to replicate this finding. We found no group effects of tDCS, as in the original study, but we no longer found a significant negative correlation. We present a series of statistical measures of replication success, all of which confirm that both studies are not in agreement. First, the correlation here is significantly smaller than a conservative estimate of the original correlation. Second, the difference between the correlations is greater than expected due to sampling error, and our data are more consistent with a zero-effect than with the original estimate. Finally, the overall effect when combining both studies is small and not significant. Our findings thus indicate that the effects of lDPLFC-tDCS on the AB are less substantial than observed in our initial study. Although this should be quite a common scenario, null findings can be difficult to interpret and are still under-represented in the brain stimulation and cognitive neuroscience literatures. An important auxiliary goal of this paper is therefore to provide a tutorial for other researchers, to maximize the evidential value from null findings.

## Introduction

The *attentional blink* (AB) phenomenon clearly demonstrates that our capacity to process incoming information is easily overwhelmed. The AB occurs when two targets are embedded in a rapidly presented stream of distractors [1] (for reviews, see [2, 3]). The first target (T1) is

available on GitHub (and also from our OSF page), in the form of an R notebook detailing all the analyses that we ran for this project, along with the results. We also include an Rmarkdown (Xie, Allaire, & Grolemund, 2018) source file for this paper that can be run to reproduce the pdf version of the text, along with all the figures and statistics. See: https://doi.org/10.5281/zenodo.4782446.

**Funding:** This work was supported by a Research Talent grant from the Netherlands Organization for Scientific Research (NWO) to HS and KR.

**Competing interests:** The authors have declared that no competing interests exist.

usually reported with little difficulty. When there is a longer lag between the two targets ($>$ 500 ms; [4]), accuracy for the second target (T2) can be on par with the first. However, for shorter lags, T2 is most often missed—as if the attentional system momentarily faltered ("blinked").

While the AB might seem to be a fundamental bottleneck, it can under some circumstances be overcome. For example, the size of the AB can be reduced by distracting activities [5–7], or after following an intensive mental training program [8]. Others have tried to use non-invasive brain stimulation [9] as a means to influence the AB. Several studies have shown that repetitive transcranial magnetic stimulation (TMS) can improve target perception in AB tasks [10–12]. Yet, as TMS did not show a differential effect for targets presented at shorter or longer lags, it did not affect the AB itself.

Transcranial direct current stimulation (tDCS) is another brain stimulation technique that has gained traction in the past two decades. In tDCS, an electrical current flows between an anodal and cathodal electrode, which can affect the excitability of the underlying cortex [13]. Anodal stimulation generally enhances cortical excitability, while cathodal stimulation may have an inhibitory effect [14, 15] (though note that this does not hold in all cases [16], and the underlying physiology is complex [17–19]).

A previous study by our group [20] recently examined the effects of tDCS on the AB. In this study, anodal and cathodal tDCS were applied over the left dorsolateral prefontal cortex (lDPLFC, with the other electrode on the right forehead)—one of the core brain areas implicated in the AB [21, 22]. At the group level, tDCS did not appear to have any effects on AB size. However, anodal and cathodal tDCS did appear to systematically affect the AB within individuals, as their effects were negatively correlated. For a given individual, this negative correlation implies that when AB size decreased (compared to a baseline measurement) during anodal tDCS, AB size would increase during cathodal tDCS (or vice versa).

Our previous findings [20] complement earlier literature showing large individual differences in both the AB [23] and effects of tDCS [24]. However, only one other tDCS study of the AB exists to date, which in fact did find a group-level effect of stimulation, in contrast to [20, 25]). Also, the negative correlation between the effects of anodal and cathodal tDCS has so far only been reported once [20], and was based on an exploratory analysis. We thus decided to conduct another study aiming to replicate this finding. Replication of findings in general is important to establish credibility of scientific claims [26], and in this specific case, reproduction of our original findings would be informative about the ability of tDCS to modulate the apparent bottleneck in information processing captured by the AB, and about the importance of left dorsolateral prefrontal cortex in the AB and attentional filtering more generally. Thus, we aimed to confirm our previous results and conclusions.

We also aimed to examine the neurophysiological mechanisms underlying the individual differences in response to tDCS using EEG [24, 27, 28]. However, since we did not replicate the behavioral effect of tDCS on the AB as in our original study [20], we were unable to fulfill this aim.

To foreshadow our results, like in our original study [20], we did not find a group effect of tDCS on the AB. However, the correlation between the effects of anodal and cathodal tDCS was also not significant. Although this indicates that the two studies may differ, a failure to reject the null hypothesis by itself does not tell us much [29]: it is crucial to also take measures of the uncertainty and effect size in both studies into account [30]. That said, there is no consensus on when a replication can be considered successful or not [31], let alone a single statistical test to definitely answer this question conclusively [32].

Therefore, we employ a number of statistical methods to maximize the evidential value in these two studies. We ask three questions (cf. [33]) that all aim to evaluate to what extent the

present study is a successful replication [34] (cf. [35, 36]) of our previous work [20]. First, while the effect in our study was not significant, there might still be a meaningful effect that is simply smaller than anticipated. Therefore, we used equivalence testing [37] to answer the question "*is the correlation in study 2 significantly small*?" Second, although our current result differs from, it could still be more consistent with our previous findings [20] than with alternative explanations. So in addition, we asked "*is the correlation in study 2 different from study 1*?" and aimed to answer this question using prediction intervals [38, 39] and replication Bayes factors [40, 41]. Finally, it could be that the effect in our study alone is not sufficiently large, but the overall effect based on both studies is. This raises the question "*is the effect significant when combining study 1 and 2*?" which we addressed through meta-analysis [42, 43] and by pooling the data.

These questions address issues of reproducibility that are currently faced by many in the brain stimulation field [44], and in the cognitive neuroscience community at large [45, 46]. Therefore, aside from our focus on tDCS and the AB, an important auxiliary goal of this article is to provide a tutorial on the statistical evaluation of replication studies (also see [35, 47]). We hope this may prove useful to other researchers who find themselves in similar situations.

## Methods

### Participants

Fourty-eight participants took part in total, eight of whom were excluded after the first session. One participant was excluded as a precaution because they developed an atypical headache after the first session, and we could not rule out this was related to the tDCS. Another stopped responding to our requests to schedule the second session. Another six participants were excluded because their mean T1 accuracy in the first session was too low, which implies they were not engaged in the task in the manner we expected, and would leave fewer trials to analyze, because our T2 accuracy measure included only trials in which T1 was seen. We used a cut-off of 63% T1 accuracy as an exclusion criterium, which was two standard deviations below the mean of a separate pilot study (n = 10).

This left a final sample of 40 participants (29 female, mean age = 20.94, *SD* = 2.45, range = 18–28). This sample size was determined a priori to slightly exceed the original study [20] (n = 34).

The experiment and recruitment took place at the University of Amsterdam. All procedures for this study were approved by the ethics review board of the Faculty for Social and Behavioral Sciences, and complied with relevant laws and institutional guidelines. All participants provided their written informed consent and were compensated with course credit or €10 per hour (typically €65 for completing two full sessions).

### Procedure

The study procedures were identical to our previous study [20]: participants received anodal and cathodal tDCS in separate sessions (Fig 1), which typically took place exactly one week apart (cf. minimum of 48 hours previously [20]). The time in between served to keep the sessions as similar as possible, and to minimize the risk of tDCS carry-over effects. 18 participants received anodal tDCS in the first session and cathodal tDCS in the second, and vice versa for the remaining 22 participants.

First, participants experienced the sensations induced by tDCS in a brief trial stimulation (see the tDCS section). Next, participants completed 20 practice trials of the task (see the Task section). For the main portion of the experiment, participants performed three blocks of the task (Fig 1): before tDCS (*baseline*), during anodal/cathodal tDCS (*tDCS*), and after tDCS

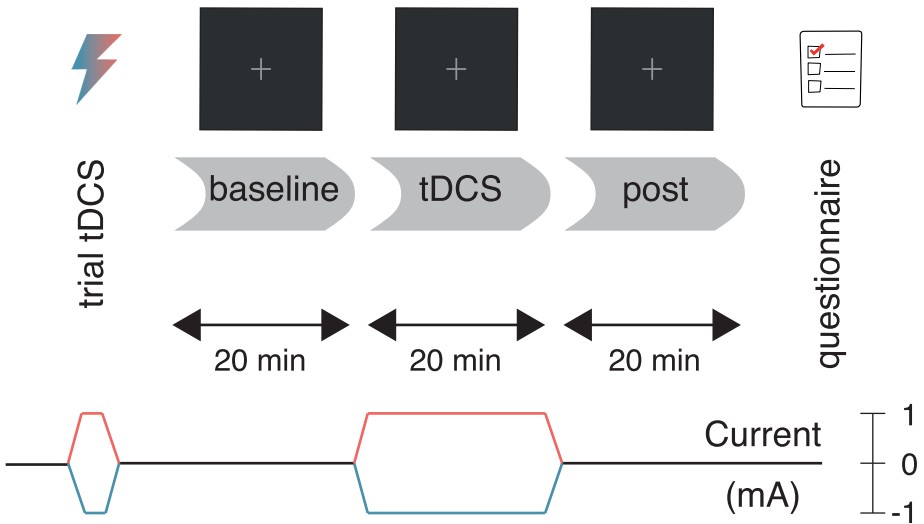

Session 1: either anodal (n=18) or cathodal (n=22) tDCS

~ 7d

Session 2: either cathodal (n=22) or anodal (n=18) tDCS

**Fig 1. Experimental design.** After a baseline block without stimulation, participants performed the attentional blink task during 20 minutes of anodal (red) or cathodal (blue) tDCS, followed by a post-test block (also without stimulation). The second session (typically 7 days later) was identical, except that the tDCS polarity was reversed.

(*post*). Finally, after completing the three blocks, participants filled in a questionnaire on tDCS-related adverse effects (see S1 Table and S1 Fig).

Within each block of the task, participants took a self-timed break every 50 trials (~5 minutes); between the blocks, the experimenter walked in. Participants performed the task for exactly 20 minutes during the *baseline* and *post* blocks. During the *tDCS* block, the task started after the 1-minute ramp-up of the current was complete, and continued for 21 minutes (constant current, plus 1-minute of ramp-down).

## Task

The attentional blink task (Fig 2) was almost identical to the one we used previously [20, 48] and (see S2 Table for a full list of the differences), which in turn was based on an earlier task [49]. A rapid serial visual presentation stream of 15 letters was shown on each trial, using Presentation software (Neurobehavioral Systems, Inc.). Each letter was displayed for 91.7 ms (11 frames at 120 Hz) on a dark gray background. The letters were presented in font size 40 (font: Courier New) at a viewing distance of 90 cm. On each trial, the letters were randomly sampled without replacement from the alphabet, excluding the letters I, L, O, Q, U and V, as they were too similar to each other. All distractor letters were mid-gray, whereas T1 and T2 were colored. T1 was red and always appeared at position 5 in the stream. T2 was green and followed T1 after either 2 distractors (*lag 3*) or 7 distractors (*lag 8*) (cf. lags 2, 4 and 10 in the original study [20]).

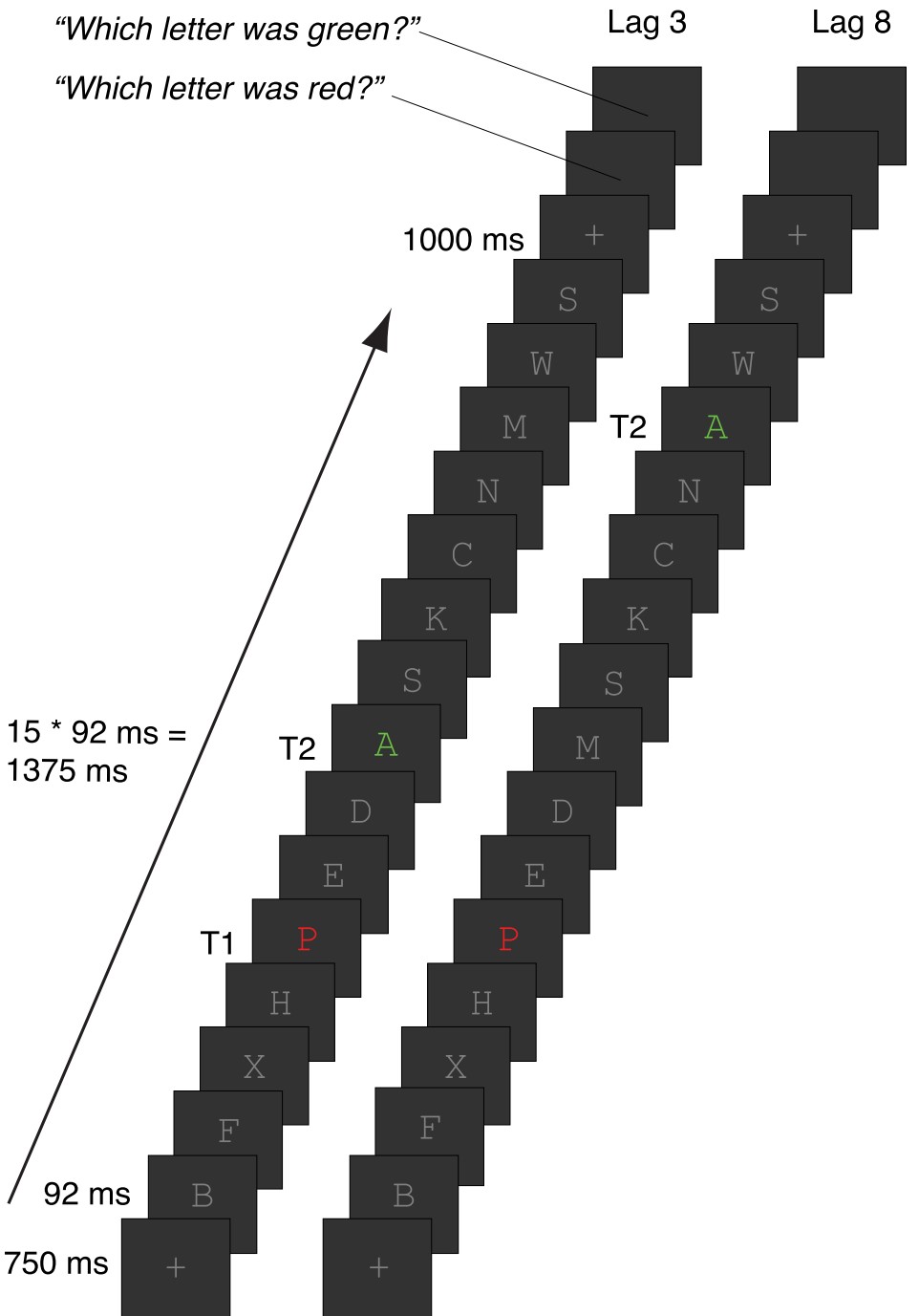

**Fig 2. Attentional blink task.** Participants viewed rapid serial visual presentation streams of 15 letters, all of which were distractors (gray letters) except for T1 and T2. T1 was presented in red at position 5; T2 was presented in green and followed T1 after 2 distractors (lag 3, inside the AB window) or 7 distractors (lag 8, outside the AB window). At the end of the trial, participants reported the identity of T1 and then T2 (self-paced).

The letter stream was preceded by a fixation cross (same color as the letters) presented for 1750 ms and followed by another fixation cross. Finally, the participant was prompted to type in (using a standard keyboard) the letter they thought was presented as T1 ("Which letter was red?"), followed by T2 ("Which letter was green?").

Trial duration varied slightly because both the T1 and T2 response questions were self-paced, so some participants completed more trials than others depending on their response times. On average, participants completed 130 short lag trials ($SD$ = 17; range = 78–163) and 65 long lag trials ($SD$ = 9; range = 39–87) per 20-minute block.

## tDCS

Transcranial direct current stimulation was delivered online (i.e. during performance of the attentional blink task) using a DC-STIMULATOR PLUS (NeuroCare Group GmbH). The current was ramped up to 1 mA in 1 minute, stayed at 1 mA for 20 minutes, and was ramped down again in 1 minute.

One electrode was placed at F3 (international 10–20 system) to target the lDLPFC; the other was placed over the right forehead, centered above the eye (approximately corresponding to position Fp2). Both electrodes were 5 x 7 cm in size (35 cm$^2$), leading to a current density of 0.029 mA/cm$^2$. The montage and tDCS parameters are identical to our previous study [20], the only exception being the conductive medium. We used Ten20 conductive paste (Weaver and Company), because it was easier to apply concurrently with the EEG equipment (see the EEG section); originally [20], saline solution was used as a conductive medium, together with rubber straps to keep the electrodes in place.

Participants received either anodal tDCS (anode on F3, cathode on right forehead) or cathodal tDCS (cathode on F3, anode on right forehead) in separate sessions. The procedure was double-blinded: both the participant and the experimenters were unaware which polarity was applied in a given session. The experimenter loaded a stimulation setting on the tDCS device (programmed by someone not involved in data collection), without knowing whether it was mapped to anodal or cathodal tDCS. In the second session, the experimenter loaded a second setting mapped to the opposite polarity (half the dataset), or simply connected the terminals of the device to the electrodes in the opposite way.

At the start of the experiment, participants received a brief trial stimulation, based on which they decided whether or not they wanted to continue with the rest of the session. The experimenter offered to terminate the experiment in case tDCS was experienced as too uncomfortable, but none of the participants opted to do so. For the trial stimulation, the current was ramped up to 1 mA in 45 seconds, stayed at 1 mA for 15 seconds, and was ramped down again in 45 seconds.

## EEG

We also recorded EEG during all three task blocks. As noted in the Introduction, we refrained from further analysis of the EEG data, given the absence of a behavioral effect. Instead, we are making the EEG data publicly available at this time on the OpenNeuro platform: https://doi.org/10.18112/openneuro.ds001810.v1.1.0. The dataset is formatted according to the Brain Imaging Data Structure (BIDS) standard [50] for EEG [51], to facilitate re-use. We include the raw data, as well as the fully preprocessed data and the MATLAB code that generated it.

## Data analysis software

Data were analyzed using R [Version 3.6.2; 52] from within RStudio [Version 1.1.463; 53]. In addition to the packages cited in the follow sections, we used the following R-packages: *BayesFactor* [Version 0.9.12.4.2; 54], *broom* [Version 0.7.4.9000; 55], *cowplot* [Version 1.1.1; 56], *emmeans* [Version 1.5.4; 57], *here* [Version 1.0.1; 58], *kableExtra* [Version 1.3.2; 59], *knitr* [Version 1.31; 60], *papaja* [Version 0.1.0.9997; 61], *psychometric* [Version 2.2; 62], *pwr* [Version 1.3.0; 63], and *tidyverse* [Version 1.3.0; 64].

## Group-level analysis

Repeated measures ANOVAs were conducted on T1 accuracy (percentage of trials where T1 was reported correctly) and T2|T1 accuracy (percentage of trials where T2 was reported correctly, out of the subset of T1-correct trials) using the *afex* package [Version 0.28–1; 65]. The same factors were included for both repeated measures ANOVAs, (following our previous approach [20]): Lag (3, 8), Block (baseline, tDCS, post), Stimulation (anodal, cathodal), and the between-subject factor Session order (anodal tDCS in the first session vs. cathodal tDCS in the first session). Effect sizes are reported as generalized eta-squared ($\hat{\eta}^2_G$) [66]. Greenhouse-Geisser-adjusted degrees of freedom ($df^{GG}$) and p-values are reported as a correction for sphericity violations.

## Individual differences analysis

We reproduced the analysis behind Fig 4 of London & Slagter [20], which showed a differential effect of anodal vs. cathodal tDCS at the individual participant level. First, we calculated AB magnitude by subtracting T2|T1 accuracy at lag 3 from T2|T1 accuracy at lag 8. Next, change scores were created by subtracting AB magnitude in the *baseline* block from the *tDCS* and the *post* blocks, respectively. The change scores in the anodal and cathodal session were then correlated to each other. Again following our previous approach [20], we computed a partial correlation (using the *ggm* package [Version 2.5; [67]]), attempting to adjust for variance due to Session order.

## Replication analyses

In contrast to [20], the analysis described in the previous section did not produce a significant correlation in our dataset. Therefore, we conducted five follow-up analyses that aim to quantify to what extent our results (do not) replicate [20]. These all provide a complementary perspective on this question. First, we performed an equivalence test (1) to assess whether the effect in the present study was significantly smaller than in our previous study [20]. While this procedure is more focused on hypothesis testing, we also constructed prediction intervals (2) to capture the range of effect sizes we can expect in a replication of our previous study [20]. Both of these procedures are based on frequentist statistics, which cannot directly quantify the amount of evidence for a (null) hypothesis. Therefore, we also computed a replication Bayes factor that expresses whether the data in the present study are more likely under the null hypothesis that the effect is absent, vs. the alternative hypothesis that the effect is comparable to the previous estimate [20]. Finally, we directly combined both studies and estimated the size of the overall effect, through meta-analysis (4) of both correlations, and by computing a new correlation on the pooled dataset (5). More details on each analysis can be found in the following sections, and the provided online resources (see the Data, materials, and code availability section).

**Equivalence tests.** Equivalence tests can be used to test for the *absence* of an effect of a specific size (see [37] for a tutorial). Usually, the effect size used for the test is the smallest effect size of interest (the SESOI). Typically, equivalence tests are two one-sided tests: one test of the null hypothesis that the effect exceeds the upper equivalence bound (positive SESOI), and one that the effect exceeds the lower equivalence bound (negative SESOI). However, a one-sided test is more appropriate here: previously we found [20] that the effects of anodal and cathodal tDCS were anticorrelated, so we are only interested in negative effect sizes. This is known as an inferiority test [37].

We follow the "small telescopes" [30] approach to set the SESOI to $r_{33\%}$: the correlation that the original study [20] had 33% power to detect. The reasoning behind this approach is that it is difficult to prove that an effect does not exist at all, but easier to show that it is surprisingly small. An equivalence test can suggest that the effect is unlikely to exceed $r_{33\%}$, such that the odds to detect it were stacked at least 2:1 against the original study [20]. That would not mean the effect does not exist at all, but it would mean the original evidence for the effect is not very convincing, as "too small a telescope" (in this case, an inadequate sample size) was used to reliably detect it.

There are many possible specifications of the SESOI, none of which are necessarily wrong or right [37]. We favored the "small telescopes" approach because it constitutes a relatively strict test—$r_{33\%}$ is much smaller than the original correlation [20]. Because the originally observed correlation [20] could have overestimated the true correlation, it is prudent to set the SESOI to be smaller. Furthermore, the approach was specifically designed to evaluate replication results [30], and has been used previously in large-scale replication studies [e.g. 35]. However, recent simulation studies have also demonstrated that equivalence tests for replications can be either too liberal (especially in the case of publication bias) [68], or too conservative (especially in case of low sample sizes) [69].

We conducted an inferiority test using the *TOSTER* package [Version 0.3.4; 70] against the null hypothesis that the correlation coefficient in the present study is at least as negative as $-r_{33\%}$. At a standard alpha level of 0.05, the test is significant if the 90% confidence interval around the observed correlation does not contain $r_{33\%}$. This would mean that the observed correlation should be considered "statistically inferior," as it is then significantly smaller (i.e. less negative) than $-r_{33\%}$.

**Prediction interval.** Prediction intervals contain a range of values we can expect a new observation to fall within. In our case, the observation of interest is the correlation between the effects of anodal and cathodal tDCS. This correlation is estimated based on a sample, and is thus subject to sampling error: any two estimates of the correlation will almost never be exactly the same. Prediction intervals aim to quantify how dissimilar two estimates can be before we should be surprised.

Here, we construct a prediction interval around the original estimate of the correlation [20]. This prediction interval contains the range of correlation coefficients we can expect to see in the present study, given the results of the original study [20]. The width of the interval depends on the sample sizes of both studies, as larger samples will reduce variation in the estimates, leading to smaller prediction intervals [39].

If the original study were replicated 100 times, 95 of the observed correlation coefficients would fall within the 95% prediction interval [39]. Note that this definition is related to, but different from, a *confidence interval*, which quantifies uncertainty about the (unknown) true correlation in the population (95 out of every hundred constructed 95% confidence intervals contain the true population parameter). Because prediction intervals concern the next single observation, they make a more specific claim, and will be wider than confidence intervals.

We calculated a 95% prediction interval for correlations, following [38], using the *predictionInterval* package [Version 1.0.0; 71].

**Replication bayes factor.** Bayes factors can be used to express the relative evidence for the null ($H_0$) or alternative hypothesis ($H_1$) [72]. In a default Bayesian hypothesis test, $H_0$ states the effect size is absent (i.e. exactly zero); $H_1$ states that the effect is present (specified further by a prior distribution of effect sizes).

In a replication context, we want to decide between two different scenarios [41]. $H_0$ is the hypothesis of an idealized skeptic, who disregards the information from the original study and believes the effect is absent. The alternative hypothesis $H_r$ belongs to an idealized proponent,

who believes that the effect is exactly as in the original study, i.e. their prior distribution is simply the posterior distribution of the original study.

We used a replication Bayes factor test for correlations [40]. The replication Bayes factor $BF_{0r}$ expresses evidence for $H_0$: "the correlation is 0" relative to $H_r$: "the correlation is as in the original study." We use the following interpretation scheme [73], where $1<BF_{0r}<3$ consitutes "anecdotal evidence" for $H_0$, $3<BF_{0r}<10$ ~ "moderate evidence," and $10<BF_{0r}<30$ ~ "strong evidence."

**Meta-analysis.** The outcomes from multiple studies on the same phenomenon can be combined through meta-analysis. Here we compute a meta-analytic estimate of the correlation based on the correlations observed here and as observed originally [20], using the *metafor* package [Version 2.4–0; [74]]. We weighed the estimate by sample size, as larger studies allow for more precise effect size estimates, and thus should also contribute more to the overall meta-analytic estimate. In our case, this means the present study will have a slightly higher influence on the meta-analytic effect size (because its sample size exceeds our previous study [20]). We specified the meta-analysis as a fixed-effects model, because both studies are highly similar and from the same population (e.g., the experiments were conducted in the same location, and the sample was from the same university student population). With a fixed-effects analysis, we estimate the size of the effect *in the set of available studies*, meaning our inferences cannot generalize beyond. A random-effects meta-analysis would be appropriate in case the studies were more dissimilar, and if we sought an estimate of *the true effect in the population*, but we would need more than just two studies for this approach. When more studies are available, and it is not immediately clear whether fixed- or random-effects meta-analysis is more appropriate, it is advisable to report both [43]. Note that while meta-analyses are a powerful way to assess the overall effect in a series of studies, they are particularly vulnerable to false positives when there is publication bias [68], or if the selection of studies (or any single study) is biased [75].

**Pooling the data.** Another approach is to pool the single-subject data from both studies, and to re-calculate the partial correlation on the combined sample (n = 74). The main difference between the two studies is that we originally [20] presented T2 at lags 2, 4 and 10; here we used lags 3 and 8. The long lags (lag 8 vs. lag 10) should be fairly comparable, as they are both well outside the attentional blink window (> 500 ms following T1; [4]). However, there should be a sizeable performance difference at the short lags (lag 2 vs. lag 3), as the attentional blink is larger at lag 2 than lag 3. Therefore, we opted to also create a "lag 3" condition in the previous data [20], by averaging T2|T1 accuracy at lag 2 and lag 4. The difference from lag 2 to 4 (and 4 to 10) in the original study [20] looks fairly linear (see Fig 3 [20]), so this seems a fair approximation of "true" lag 3 performance. Afterwards we recomputed the partial correlation between AB magnitude change scores as described previously (see the Individual differences analysis section).

Note that this analysis is tailored to this series of studies, and not generally advisable. To get a more accurate estimate of the effect at lag 3, it is necessary to redo the analysis on the larger, combined sample. But repeating a statistical test after collecting more data ("optional stopping") invalidates the interpretation of the p-value and can drastically increase the false positive rate [76]. This would only be acceptable when the analysis plan has been preregistered, and the false positive rate of sequential analyses is controlled (for potential solutions, see [77]). We therefore do not report a p-value for this test, but only the effect size and its confidence interval.

## Data, materials, and code availability

All of the data and materials from this study and the data from the original study [20] are available on the Open Science Framework: https://doi.org/10.17605/OSF.IO/Y6HSF. The analysis

code is available on GitHub (https://doi.org/10.5281/zenodo.4782446; and also from our OSF page), in the form of an R notebook detailing all the analyses that we ran for this project, along with the results. We also include an Rmarkdown [78] source file for this paper that can be run to reproduce the pdf version of the text, along with all the figures and statistics.

# Results

## Group-level analysis

Fig 3 shows the attentional blink (T2|T1 accuracy per lag) for each of the three blocks (pre, tDCS, post) and stimulation conditions separately. The summary statistics and ANOVA results for T2|T1 accuracy can be found in Tables 1 and 2. There was a clear attentional blink effect on average (main effect of Lag, $F[1,38] = 432.11$, $p<.001$), as T2|T1 accuracy for lag 8 was higher than lag 3.

Effects of tDCS at the group-level should manifest as a three-way interaction between Block, Stimulation, and Lag. As in the original study [20], this interaction was not significant ($F[1.77,67.17] = 2.77$, $p = .076$). However, the higher-order interaction with Session order did reach significance ($F[1.77,67.17] = 7.25$, $p = .002$). This interaction appears to be mostly driven by a learning effect across sessions and blocks (the Session order by Stimulation by Lag interaction). From the first to the second session, lag 3 performance improved, while lag 8 performance declined somewhat (leading to a smaller attentional blink). This change was stronger in participants that received anodal tDCS in the first session, and less pronounced in the cathodal-first group. We do not consider this a genuine effect of tDCS on the attentional blink, because there is no clear reason why these randomized groups should differ, and because the largest difference between the anodal and cathodal session occurred in the baseline block already (see Table 1).

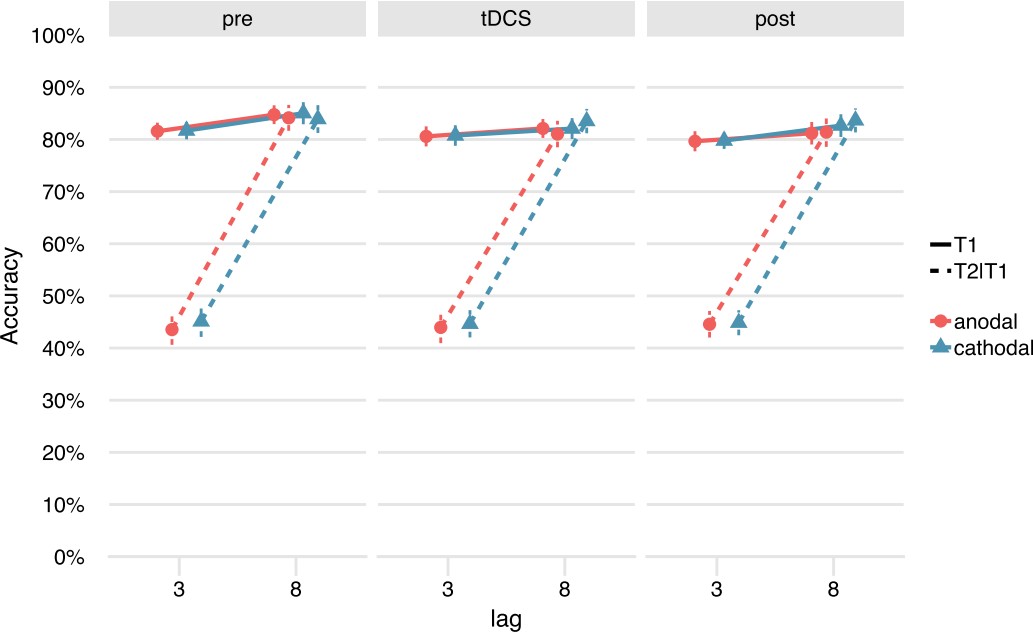

**Fig 3. No effects of tDCS on the attentional blink at the group level.** There was a clear attentional blink effect: a lower % T2 accuracy (given T1 correct: T2|T1; dashed lines) for lag 3 (T2 presented inside the attentional blink window) than lag 8 (T2 presented outside the attentional blink window, on par with T1 accuracy). However, the attentional blink did not change systematically over stimulation conditions (anodal, cathodal) and blocks (pre, tDCS, post). T1 accuracy (solid lines) was also not affected by tDCS.

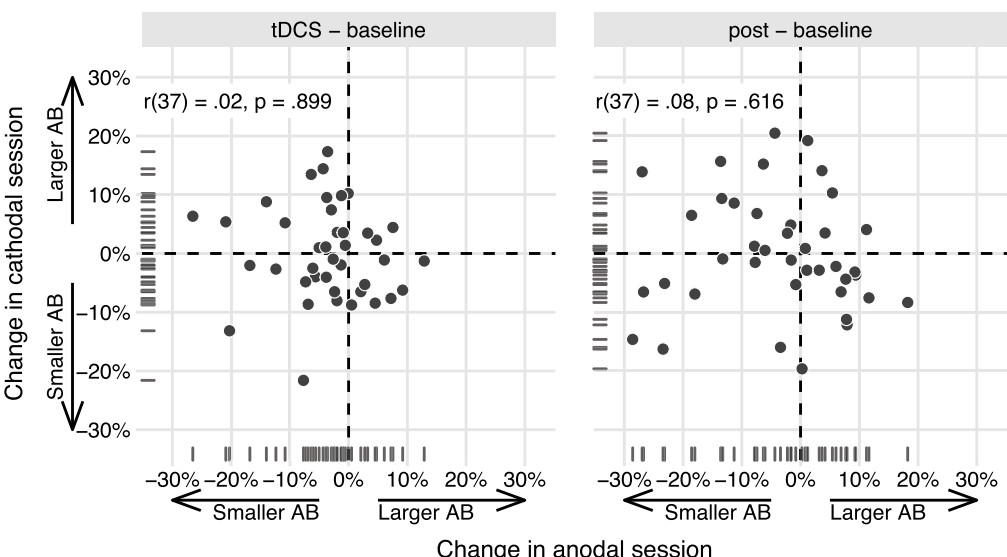

**Fig 4. The effects of anodal and cathodal tDCS are not correlated in the present study.** Data points show AB magnitude change scores (tDCS—baseline, post—baseline) for each participant in the study, in the anodal session (x-axis) and the cathodal session (y-axis). While we originally [20] found a negative partial correlation (for tDCS—baseline), suggesting opposite effects of anodal and cathodal tDCS, this effect appears to be absent here. The partial correlation coefficients (attempting to adjust for Session order) and p-values are printed in the upper left.

We ran the same repeated measures ANOVA for T1 accuracy (Table 3). On average, participants correctly identified T1 in 82% of trials (Fig 3), which is comparable to previous studies using this task (86% [20], 82% [48]). T1 accuracy was also slightly lower for lag 3 than lag 8 (main effect of Lag, $F[1,38] = 29.23$, $p < .001$). There was a main effect of Block, reflecting that T1 accuracy decreased *within* a session ($F[1.89,71.87] = 6.64$, $p = .003$). Finally, T1 performance also decreased *across* the sessions (interaction between Session order and Stimulation: $F[1,38] = 11.24$, $p = .002$).

In sum, we conclude that there is no significant effect of tDCS on the attentional blink or T1 accuracy at the group level, in agreement with our previus study [20].

## Individual differences

Our main aim was to replicate our original work [20], that revealed a negative correlation between AB magnitude change scores (comparing the *tDCS* and *baseline* blocks) in the anodal

**Table 1. Descriptive statistics for T2|T1 accuracy.**

| | | First session: anodal (n = 18) | | First session: cathodal (n = 22) | |
|---|---|---|---|---|---|
| | | anodal | cathodal | anodal | cathodal |
| baseline | | | | | |
| | lag 3 | 0.395 (0.149) | 0.486 (0.167) | 0.469 (0.211) | 0.423 (0.168) |
| | lag 8 | 0.826 (0.119) | 0.819 (0.095) | 0.854 (0.125) | 0.855 (0.087) |
| tDCS | | | | | |
| | lag 3 | 0.415 (0.168) | 0.450 (0.165) | 0.460 (0.201) | 0.444 (0.188) |
| | lag 8 | 0.787 (0.131) | 0.819 (0.098) | 0.830 (0.115) | 0.849 (0.096) |
| post | | | | | |
| | lag 3 | 0.437 (0.171) | 0.451 (0.160) | 0.453 (0.190) | 0.447 (0.164) |
| | lag 8 | 0.783 (0.118) | 0.825 (0.098) | 0.840 (0.145) | 0.846 (0.120) |

Values are "Mean (SD)".

**Table 2. Repeated measures ANOVA on T2|T1 accuracy.**

| Effect | F | $df_1^{GG}$ | $df_2^{GG}$ | p | $\hat{\eta}_G^2$ |
|---|---|---|---|---|---|
| Session order | 0.33 | 1 | 38 | .568 | .006 |
| Block | 1.13 | 1.91 | 72.71 | .328 | .001 |
| Stimulation | 2.47 | 1 | 38 | .125 | .002 |
| Lag | 432.11 | 1 | 38 | < .001 | .634 |
| Session order × Block | 0.29 | 1.91 | 72.71 | .741 | .000 |
| Session order × Stimulation | 5.55 | 1 | 38 | .024 | .005 |
| Session order × Lag | 0.48 | 1 | 38 | .494 | .002 |
| Block × Stimulation | 0.28 | 1.91 | 72.51 | .747 | .000 |
| Block × Lag | 1.67 | 1.70 | 64.73 | .199 | .001 |
| Stimulation × Lag | 0.10 | 1 | 38 | .751 | .000 |
| Session order × Block × Stimulation | 1.93 | 1.91 | 72.51 | .154 | .001 |
| Session order × Block × Lag | 0.24 | 1.70 | 64.73 | .752 | .000 |
| Session order × Stimulation × Lag | 5.84 | 1 | 38 | .021 | .002 |
| Block × Stimulation × Lag | 2.77 | 1.77 | 67.17 | .076 | .001 |
| Session order × Block × Stimulation × Lag | 7.25 | 1.77 | 67.17 | .002 | .003 |

and cathodal sessions ($r(31) = -.45$, 95%CI[-0.68, -0.12], $p = .009$). Their interpretation was that the effects of anodal and cathodal tDCS were anti-correlated: those participants that improve their performance (i.e., show a smaller AB) due to anodal tDCS tend to worsen due to cathodal tDCS, and vice versa.

We ran the same partial correlation (attempting to adjust for Session order) between the anodal and cathodal AB magnitude change scores (*tDCS—baseline*) on our data (Fig 4). However, here the resulting correlation did not go in the same direction ($r(37) = .02$, 95%CI[-0.30, 0.33]), and was not significant ($p = .899$). Or, adopting the terminology suggested to describe replications [79]: we obtained *no signal* (the 95% confidence interval includes 0, and therefore the correlation is not significant), and our result is *inconsistent* (the 95% confidence interval excludes the point estimate of the original correlation [20]). In the next sections, we present a

**Table 3. Repeated measures ANOVA on T1 accuracy.**

| Effect | F | $df_1^{GG}$ | $df_2^{GG}$ | p | $\hat{\eta}_G^2$ |
|---|---|---|---|---|---|
| Session order | 0.96 | 1 | 38 | .332 | .018 |
| Block | 6.64 | 1.89 | 71.87 | .003 | .010 |
| Stimulation | 0.00 | 1 | 38 | .996 | .000 |
| Lag | 29.23 | 1 | 38 | < .001 | .013 |
| Session order × Block | 0.60 | 1.89 | 71.87 | .540 | .001 |
| Session order × Stimulation | 11.24 | 1 | 38 | .002 | .030 |
| Session order × Lag | 0.04 | 1 | 38 | .844 | .000 |
| Block × Stimulation | 0.24 | 1.92 | 73.08 | .777 | .000 |
| Block × Lag | 1.91 | 1.93 | 73.20 | .158 | .001 |
| Stimulation × Lag | 0.31 | 1 | 38 | .584 | .000 |
| Session order × Block × Stimulation | 9.94 | 1.92 | 73.08 | < .001 | .007 |
| Session order × Block × Lag | 0.19 | 1.93 | 73.20 | .821 | .000 |
| Session order × Stimulation × Lag | 0.96 | 1 | 38 | .333 | .000 |
| Block × Stimulation × Lag | 0.31 | 1.86 | 70.75 | .718 | .000 |
| Session order × Block × Stimulation × Lag | 0.53 | 1.86 | 70.75 | .580 | .000 |

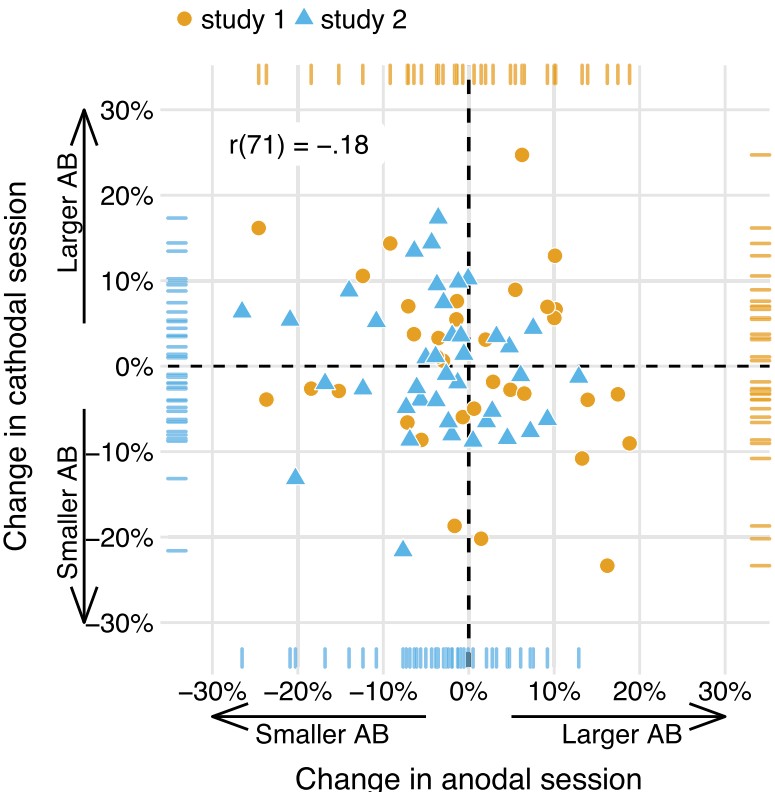

**Fig 5. The effects of anodal and cathodal tDCS are also not correlated when pooling data from both studies.** As in Fig 4, data points show AB magnitude change scores (tDCS—baseline) in the anodal session (x-axis) and the cathodal session (y-axis), but now for each participant in study 1 [20] and the present study (study 2). While study 1revealed a negative partial correlation, suggesting opposite effects of anodal and cathodal tDCS, this effect appears to be absent when based on the combined data of both studies. The partial correlation coefficient (attempting to adjust for Session order) is printed in the upper left.

number of follow-up analyses that further explore this difference in results between both studies (see the Replication analyses subsection in the Methods section).

**Is the correlation in study 2 significantly small?.** We originally [20] found a medium- to large correlation ($r = -.45$), but the correlation we find here is much smaller ($r = .02$). We use equivalence testing ([37]; see the Equivalence tests subsection in the Methods section) to assess whether this correlation is significantly smaller than a smallest effect size of interest (SESOI). Following the "small telescopes" approach [30], we set the SESOI to $r_{33\%}$, the correlation the original study [20] had 33% power to detect. Given their sample size of 34 participants, $r_{33\%}$ = .27.

An inferiority test shows that the correlation here is significantly less negative than $-r_{33\%}$ ($p = .038$) (Fig 6), although only by a small margin. The effect is therefore "statistically inferior": the correlation does not exceed the lower equivalence bound ($-r_{33\%}$ = -.27). By this definition, the correlation is too small (i.e. not negative enough) to be considered meaningful, indicating that we did not successfully replicate the previous study [20].

**Is the correlation in this study (study 2) different from the previous study (study 1)?.** To evaluate whether the correlation in the present study (study 2) was to be expected based on study 1 [20], we constructed a 95% *prediction interval* (PI) [38], using the correlation in study 1 and the sample size of both studies (see the Prediction interval subsection of the Methods section).

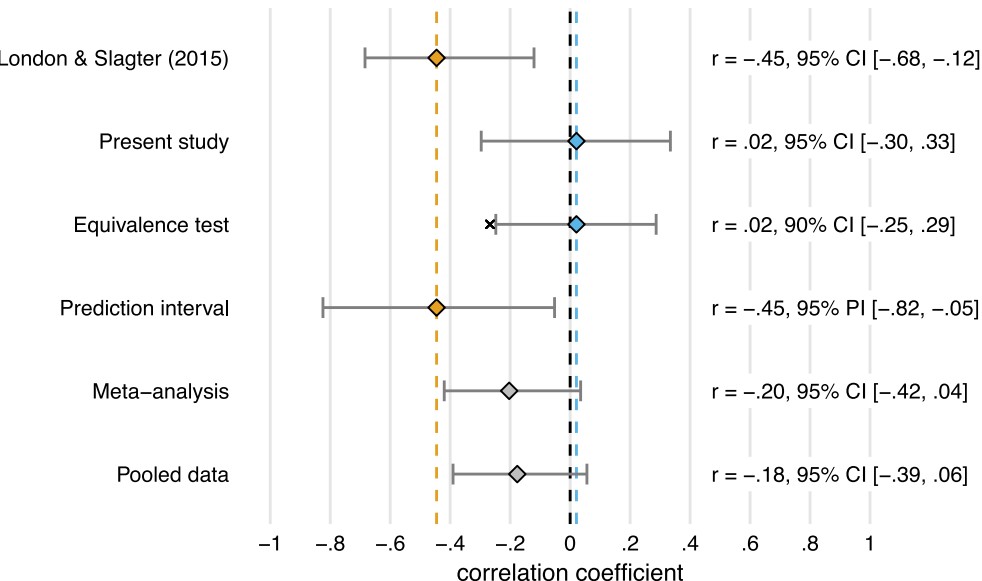

**Fig 6. Summary of all the replication analyses (with exception of the replication bayes factor).** The first two rows show the partial correlation (attempting to adjust for Session order) between the AB magnitude change scores (tDCS—baseline) in the anodal and cathodal sessions, for study 1 [20] (in yellow) and the present study (in blue). The first is significantly negative, the second is slightly positive and not significant, because its 95% confidence interval (CI) overlaps with zero. The third row shows the 90% CI around the correlation in the present study. Because this interval does not overlap with the "small telescopes" effect size, (indicated by the x: $-r_{33\%} = -.27$), this correlation is significantly smaller. The fourth row shows the 95% prediction interval (PI) around the correlation in [20]. Because this interval does not overlap with the correlation in the present study, both correlations are not consistent. The final two rows show the overall effect when the two correlations are meta-analyzed, and when one correlation is computed over the pooled data from both studies. Neither are significant (95% CI overlaps with zero). Thus, our replication analyses all suggest that we failed to replicate London & Slagter [20], and when the results are examined in combination, no evidence in support of a negative relationship between the AB magnitude change scores (tDCS—baseline) in the anodal and cathodal sessions is obtained.

The 95% PI[-0.82, -0.05] around the original correlation ($r$ = -.45) is very wide, so almost any negative correlation would fall within it. However, it does not include the replication result ($r$ = .02) (Fig 6). Assuming the results of both studies differ only due to sampling error, the correlation observed here has a 95% chance to fall within the interval. This means the correlation in our replication study is inconsistent with the correlation in the original study [20], and thus that the replication should not be considered successful.

Another approach to quantify the difference between study 1 and 2 is to construct a replication Bayes Factor [40]. We use this Bayes factor to assess evidence for $H_0$: "the effects of anodal and cathodal tDCS are uncorrelated" relative to $H_r$: "the effects of anodal and cathodal tDCS are anticorrelated [20]" (see the Replication Bayes factor subsection of the Methods section).

This replication Bayes factor expresses that the data are $BF_{0r}$ = 9.66 times more likely under $H_0$ than under $H_r$. This constitutes moderate to strong evidence that the effect is absent vs. consistent with the original [20], and thus that the effect did not replicate.

**Is the effect significant when combining study 1 and 2?.**   The sample size in both study 1 (n = 34) and 2 (n = 40) is on the lower end. Therefore, we might get a more accurate estimate of the size of the effect when combining both studies.

The meta-analytic estimate of the correlation is $r$ = -0.20, 95%CI[-0.42, 0.04], $p$ = .094 (Fig 6). So, when combining the original correlation from [20] and the correlation observed here, the overall effect is no longer significant.

In addition, we pooled the data from both studies at the single-subject level and re-calculated the partial correlation on the combined sample (n = 74). To make the datasets as comparable as possible, we first averaged over T2|T1 accuracy at lags 2 and 4 in the previous study [20], to mimic a "lag 3" condition cf. the present study (Fig 5). The resulting T2|T1 accuracies at lag 3 are very comparable across the two studies (study 1: $M$ = 0.65, SD = 0.26; study 2: $M$ = 0.64, SD = 0.24).

The partial correlation based on the pooled data is $r(71)$ = -.18. Thus, the correlation across a combination of both samples is a lot smaller than in the first study [20], and slightly smaller still than the meta-analytic estimate that included the original correlation from the first study (Fig 6).

In sum, all the analyses we conducted seem to support the same conclusion: that the current study is not a successful replication of the previous [20], and that therefore the purported opposing effect of anodal and cathodal tDCS on the attentional blink within individuals is called into question.

## Discussion

Our recent study [20] was the first to uncover potential individual differences in the effect of tDCS on the AB. This original study revealed a negative correlation between the effects of anodal and cathodal tDCS on T2|T1 accuracy across individuals (in the absence of a group-level effect of tDCS). This analysis suggested an interesting pattern of individual differences in response to tDCS: those that tended to benefit from anodal tDCS (i.e., whose AB became smaller compared to baseline) would tend to worsen during cathodal tDCS, and vice versa. This finding seemed plausible given the substantial and well-documented individual differences in both the AB [23] and the effects of tDCS [24].

We conducted a replication study, and again found no effect of tDCS at the group level. However, in contrast to our prior work [20], the correlation between the AB magnitude change scores (tDCS—baseline) in the anodal and cathodal sessions here is not significant, and not in the same direction. We therefore also computed several statistical measures of replication success focused on the negative correlation between anodal and cathodal tDCS. As discussed in more detail below, the field of electrical brain stimulation is characterized by inconsistent findings across studies. It is hence important that researchers in this field also start to incorporate methods that permit quantification of replication success. It is our hope that the overview of main, complementary analyses that researchers can use to maximize the evidential value from negative findings, provided here, will serve as a guideline for future studies.

The measures of replication success showed that in our study the observed correlation is smaller than we found previously [20], and than the smallest correlation they could have reasonably detected (i.e., an equivalence test for the lower bound of $r_{33\%}$ [30] was significant). The difference between the two studies is greater than expected based on sampling error alone (i.e., the correlation in the present study falls outside of the 95% prediction interval). Further, the data provide moderate to strong evidence for the null hypothesis of zero correlation compared to the alternative that the correlation is as in the original study [20] (i.e., we found a replication Bayes factor in favor of the null hypothesis of ~10). Finally, combining both studies yields a smaller and non-significant correlation (both in a meta-analysis and by pooling the data).

The overall picture is consistent: all measures indicate that the present study is not a successful replication of our original study [20]. But because both studies were similar in design and sample size, it is not warranted to dismiss the previous findings. Our analysis revealed substantial variation between both estimates of the effect, and so the conclusion whether tDCS is

effective should not be based on any single study, nor on only a single replication study [80]. In any series of studies, one should expect to sometimes not find a significant result, purely due to random variation—even if there is a true effect [81].

Still, the marked difference in results of both studies is surprising: they are similar enough that the present study [34] could be considered a direct replication of the first study [20]. We used the exact same electrode montage and tDCS parameters, followed the same experimental design (Fig 1), ran the experiment in the same location with the same participant population, and used a virtually identical task. Nevertheless, there are some discrepancies between the studies that could have contributed to the different outcome (see S2 Table for a full list of the differences).

The most important difference is that in the previous study [20], T2 was presented at lags 2, 4, and 10, whereas we now used lags 3 and 8. This means that AB magnitude (the difference in T2|T1 accuracy between the shortest and longest lag) was on average smaller in the present study. We introduced the change for precisely this reason: to have about as many trials for the EEG analyses in which T2 was seen vs. missed, we needed the AB to be smaller. But this increase in average T2|T1 accuracy may have also reduced the between-subject variability that is essential for analyses that capitalize on individual differences. Indeed, from Fig 5 it seems that the previous sample [20] had a larger spread, at least in the change scores for AB magnitude.

Though the change in lags is probably the most important, the concurrent EEG measurement did introduce other differences as well. Each session took longer to complete, because the EEG setup required extra time, and the pace of the task was slowed down with longer inter-stimulus intervals. Finally, the current flow could have changed due to the presence of the EEG cap and electrodes, as well as the use of conductive paste instead of saline solution.

At the group level, our results are in agreement [20]: we find no differential effects of anodal and cathodal tDCS on the AB. However, another study [25] did report a group-level effect of tDCS on the AB, but this study used different stimulation parameters and was fairly small (see our previous paper [20] for a more extensive discussion of the differences). At least with our AB task and fairly standard electrode montage and stimulation parameters [82], the AB does not appear to be very amenable to tDCS over the lDLPFC. Our null results are consistent with recent reviews and meta-analyses highlighting there is little evidence that prefrontal tDCS can be used to enhance cognition [83]; or if so, that its effects on attention [84] and cognition more generally [85] are difficult to predict, rather small [86], and restricted to a limited subset of outcome measures and stimulation parameters [87, 88].

Interpreting null results is always difficult, especially in brain stimulation studies [89]. Ultimately, a myriad of possible underlying explanations may apply, most of which we have no direct access to. For one, we cannot be sure that the current density in the lDLPFC reached sufficient levels [90, 91] based on our montage alone. More precise targeting of the stimulation towards the lDPLFC [92] and modeling of the current flow [93, 94] could provide some more confidence. Still, it would be difficult to verify that anodal tDCS and cathodal tDCS indeed had an opposite effect on cortical excitability [95, 96]. This assumption holds in many cases, and would provide a plausible explanation [24] for the negative correlation we originally found [20]. But ideally, it should be validated with direct measures of cortical excitability. Such measures are hard to obtain, although some recent studies suggest that combining tDCS with magnetic resonance spectroscopy can be used for this purpose [97–99]. Combining tDCS with MRI may also help predict individual differences in stimulation efficacy. One recent study found that individual variability in the cortical thickness of left prefrontal cortex accounted for almost 35% of the variance in stimulation efficacy among subjects [100].

It is also possible that we do not find an effect because the studies did not have sufficient statistical power [101]. Especially for the individual differences analyses, the sample size in both studies is on the lower end. For example, to detect a medium-sized correlation ($r = 0.3$) with 80% power, the required sample size is n = 84. We do approach this sample size when combining both studies (n = 74). Also, the correlation in [20] was larger ($r = -0.45$), but we cannot have much confidence in the precision of this estimate: the 95% confidence interval is so wide (-0.68 to -0.12) that it is consistent with a large range of effect sizes, and therefore almost unfalsifiable [102]. To estimate the size of a medium correlation ($r = 0.3$) within a narrower confidence band (± 0.15 with 90% confidence), a sample size of n = 134 would be required [103]. Although our analyses suggest that the correlation is small if anything, this means we cannot accurately estimate how small—even with the combined sample.

These are all decisions to be made at the design stage, which can increase the value of a null result [89]. However, after the data are in, additional tools are available to increase the value of a null result [29], especially in the case of a replication study [30], which we have demonstrated here.

We hope our paper provides inspiration for others in the fields of brain stimulation and cognitive neuroscience. Many speak of a crisis of confidence [44] and fear that the current literature is lacking in evidential value [83]. This is certainly not a unique development, as these sentiments [104] and low rates of replication abound in many research areas [35, 36, 105]. But it is perhaps aggravated by the fact that the brain stimulation field has not matured yet [106]. At the same time, this also leaves room for the field to improve, and more work is necessary to determine the potential of electrical brain stimulation [107].

To make sure the literature accurately reflects the efficacy of the technique, it is crucial to combat publication bias. Positive results are heavily over-represented in most of the scientific literature, [108–110], which has recently prompted a call to brain stimulation researchers to also publish their null results (https://www.frontiersin.org/research-topics/5535/non-invasive-brain-stimulation-effects-on-cognition-and-brain-activity-positive-lessons-from-negativ). Preregistration and registered reports are other important methods to combat publication bias that the field of electrical brain stimulation should embrace (see e.g., [111]). Publishing null results and replication studies—and making the most of their interpretation—is crucial to better this situation.

## Supporting information

**S1 Fig. tDCS adverse events.** Number of reports out of 89 sessions (either anodal or cathodal tDCS). Top row shows intensity ratings [little, moderate, strong, very strong]; bottom row shows participant's confidence that event was related to tDCS [unlikely, possibly, likely, very likely]. Adverse events are sorted in descending order of number of reports (for very rare events (five reports or fewer for a given polarity), some text counts have been removed to prevent overlap).
(PDF)

**S1 Table. Number of reports of tDCS side effects.** The intensity rating question read: "To which degree were the following sensations present during stimulation?". The confidence rating question read: "To which degree do you believe this was caused by the stimulation?".
(DOCX)

**S2 Table. Differences in the methodology and participant samples of London & Slagter [20] and the present study.**
(DOCX)

## Acknowledgments

We thank Raquel London for sharing all her experience and materials, as well as Daphne Box and Esther van der Giessen for their assistance in data collection.

## Author Contributions

**Conceptualization:** Leon C. Reteig, Heleen A. Slagter.

**Data curation:** Leon C. Reteig, Lionel A. Newman.

**Formal analysis:** Leon C. Reteig, Lionel A. Newman.

**Funding acquisition:** K. Richard Ridderinkhof, Heleen A. Slagter.

**Investigation:** Leon C. Reteig, Lionel A. Newman.

**Methodology:** Leon C. Reteig.

**Project administration:** Leon C. Reteig, Lionel A. Newman, Heleen A. Slagter.

**Resources:** Leon C. Reteig, Heleen A. Slagter.

**Software:** Leon C. Reteig.

**Supervision:** Leon C. Reteig, K. Richard Ridderinkhof, Heleen A. Slagter.

**Visualization:** Leon C. Reteig.

**Writing – original draft:** Leon C. Reteig.

**Writing – review & editing:** Leon C. Reteig, Lionel A. Newman, K. Richard Ridderinkhof, Heleen A. Slagter.

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
