## [Decision Letter · Decision Letter 0]

2 Nov 2021

PONE-D-21-17725Effects of tDCS on the attentional blink revisited: A statistical evaluation of a replication attemptPLOS ONE

Dear Dr. Slagter,

Thank you for submitting your manuscript to PLOS ONE. After careful consideration, we feel that it has merit but does not fully meet PLOS ONE’s publication criteria as it currently stands. Therefore, we invite you to submit a revised version of the manuscript that addresses the points raised during the review process.

We look forward to receiving your revised manuscript.

Kind regards,

Jorge Leite, Ph.D.

Academic Editor

PLOS ONE

Journal Requirements:

“This work was supported by a Research Talent Grant from the Netherlands 595 Organization for Scientific Research (NWO) (452-10-018, to HAS and KRR). We thank 596 Raquel London for sharing all her experience and materials, as well as Daphne Box and 597 Esther van der Giessen for their assistance in data collection.”

“This work was supported by a Research Talent grant from the Netherlands Organization for Scientific Research (NWO) to HS and KR.”

Reviewers' comments:

Reviewer's Responses to Questions

**Comments to the Author**

1. Is the manuscript technically sound, and do the data support the conclusions?

Reviewer #1: Yes

2. Has the statistical analysis been performed appropriately and rigorously? 

Reviewer #1: Yes

3. Have the authors made all data underlying the findings in their manuscript fully available?

Reviewer #1: Yes

4. Is the manuscript presented in an intelligible fashion and written in standard English?

Reviewer #1: Yes

5. Review Comments to the Author

Reviewer #1: Dear editor, thank you for the opportunity to review the manuscript “Effects of tDCS on the attentional blink revisited: A statistical evaluation of a replication attempt”. The manuscript is highly interesting and it aimed to investigate the he effects of transcranial direct current stimulation (tDCS) over the left dorsolateral prefrontal cortex on the attentional blink, precisely, attempting to replicate the findings from her previous work (London et al. [2021]). Although the authors did not succeed in replicating previous findings, one of the greatest things in the present manuscript concerns the detailed methodological description that, as the authors wrote, I also hope that provides a guideline for future studies. Please find some of my comments and suggestions below:

1) Although the abstract is clear and well-written, I personally think that it is a bit uncommon to see a reference so many times repeated in it and described in detail. Obviously, I understand that the main goal of the current manuscript is to replicate those findings, but, anyways, I would recommend changing it as really minor concern. Especially when the cited work is from the same lab, maybe a simple modification such as “in our previous work we showed that…”. So, most often along the text when the authors refer as London and Slagter (2021) It really “sounds” as another lab. I would recommend rephrasing making it clear that the same lab did both studies.

2) A minor issue. What is the ultimate purpose of “improving” AB or even why the authors want to replicate previous findings? Is there any other reason than replicating the previous results? I know that the question sounds mean at first glance, let me explain. a) I tend to guess that there might be an adaptive significance behind the AB, possibly as a filter for information processing in context, precisely to not overwhelm a cognitive system with ephemeral stimuli, i.e., to ensure conscious access to task-relevant input at the expense of task-irrelevant input. In this regard, I missed some broader statements in the introduction about the real-world possible gains of potentializing/ improving it. Furthermore, I understand that AB might be an interesting phenomenon to investigate the boundaries between low and high-level cognitive processes, such as attentional shifting/switching, or even consciousness. But what do the authors want to prove by replicating or not their previous findings? In other words, how the manuscript may enlighten on some of the current theories of attention (e.g., inhibition, inference, attention capacity, two-stage processing)? How does this brick might contribute in a building block of knowledge? In the way that the Introduction is right now I tend to think that the main goal was to provide a tutorial on the statistical evaluation of replication studies rather than the focus on tDCS and the AB. If this was the main goal, as it seems, then I would suggest to rephrase the sentence “aside from our focus on tDCS and the AB, an important auxiliary goal of this article is to provide a tutorial on the statistical evaluation of replication studies” (p.5, line 113), because this would not be an auxiliary goal, but the main one.

3) In the method, the authors wrote that “participants received anodal and cathodal tDCS in separate sessions which typically took place exactly one week apart (cf. minimum of 48 hours in London and Slagter (2021))”. Next, at some point the authors wrote “The attentional blink task was almost identical to the one used in London and Slagter (2021)”. Then, the authors wrote “The letter stream was preceded by a fixation cross (same color as the letters) presented for 1750 ms (cf. 480 ms in London and Slagter (2021)) and followed by another fixation cross (cf. none in London and Slagter (2021))”. Or the authors wrote “We used Ten20 conductive paste (Weaver and Company), because it was easier to apply concurrently with the EEG equipment (see the EEG section); London and Slagter (2021) used saline solution as a conductive medium, together with rubber straps to keep the electrodes in place.” All these sentences, and others along the text, raised some questions on what extend both procedures were, in fact, identical. Because this study is being sold as a replication study, I think that would be extremely useful to find a table in which all the main parameters as well as possible group differences between both samples are shown. A sample size of 34 or 40 participants is certainly not bad for a study using stimulation, but, due to the several experimental issues (e.g., randomization of the procedures, number of the task training trials, age, sex distribution, education level, facilities and equipment, expertise of the researcher at data collection, the fixation cross time) any minor change might really impact the outcome. For instance, in the London and Slagter (2021) the authors had 20 women out of 34 participants, as far as I understood, meaning 58%, while in the present manuscript this percentage is 72%. Therefore, I would really appreciate a complete descriptive table of both samples as well as the mean (median too, maybe) difference between both sections in the two studies and other possible confounders. The replication is not a matter of statistical analysis, only. So, I think the reader would appreciate a complete view and comparison of both research sets and participants, side-by-side. This is reinforced by the fact that neither London and Slagter (2021) and the present manuscript have a descriptive table of participants and experimental procedures such as the performance on the averaged “lag 3” in study 1 and the “lag 3” in the present study, for instance. It seems that the authors were so focused in demonstrating several methodological strategies that they underestimated very basic descriptive data on participants and procedures.

4) The entire statistical analysis is well-written and easy to follow and Figure 6 in the results section was also great add on in the manuscript. However, in figure 6 the legend with the colors is missing. Minor issue.

5) Personally, I think that both the meta-analysis and the pooled analysis were good ideas. Nevertheless, some things need some additional notes. As a minor concern, the authors wrote that to deal with different sample sizes they “weighed the estimate by sample size, so the present study will have a slightly higher influence on the meta-analytic effect size”, but no reference is reported neither a clear justification to do so and the implications of it. I am not sure about this strategy either, since the difference in sample size is not that huge and by doing so the final outcome, the effect sized obtained gets a bit trick to interpret. However, another concern has to do with not using a random effect meta-analytical procedure. Although the authors have written a very interesting and important justification in this regard, along the text the authors also made clear that both experimental designs are similar, i.e., not equal. Therefore, a random effect for study could be considered or, at least, statistically proved that it is not needed (e.g., CCI). Moreover, even if the procedures would be equal, participants are certainly not, right? So, a random effect for participants could also be considered, especially because as the authors wrote “the field of electrical brain stimulation is characterized by inconsistent findings across studies” raising the questions about individual differences and characteristics, as the authors wrote “individual variability in the cortical thickness of left prefrontal cortex accounted for almost 35% of the variance in stimulation efficacy among subjects”.

6. PLOS authors have the option to publish the peer review history of their article (what does this mean?). If published, this will include your full peer review and any attached files.

Reviewer #1: **Yes: **Dr. Bruno Kluwe-Schiavon

---

## [Author Response · Author response to Decision Letter 0]

30 Dec 2021

We thank the reviewer, Dr. Kluwe-Schiavon, for their thoughtful and constructive comments. We have revised the manuscript accordingly and feel that this has much strengthened the paper. 

1) Although the abstract is clear and well-written, I personally think that it is a bit uncommon to see a reference so many times repeated in it and described in detail. Obviously, I understand that the main goal of the current manuscript is to replicate those findings, but, anyways, I would recommend changing it as really minor concern. Especially when the cited work is from the same lab, maybe a simple modification such as “in our previous work we showed that…”. So, most often along the text when the authors refer as London and Slagter (2021) It really “sounds” as another lab. I would recommend rephrasing making it clear that the same lab did both studies.

We agree that this sounded a bit awkward and that it might not have been immediately clear that London & Slagter (2021) was co-authored by the same last author as the present study. We have kept only the first citation in the abstract and now refer explicitly to London & Slagter (2021) as previous work from our group. In the Introduction and Discussion, we have also clarified that this concerns previous work from our group.

2) A minor issue. What is the ultimate purpose of “improving” AB or even why the authors want to replicate previous findings? Is there any other reason than replicating the previous results? I know that the question sounds mean at first glance, let me explain. a) I tend to guess that there might be an adaptive significance behind the AB, possibly as a filter for information processing in context, precisely to not overwhelm a cognitive system with ephemeral stimuli, i.e., to ensure conscious access to task-relevant input at the expense of task-irrelevant input. In this regard, I missed some broader statements in the introduction about the real-world possible gains of potentializing/ improving it. Furthermore, I understand that AB might be an interesting phenomenon to investigate the boundaries between low and high-level cognitive processes, such as attentional shifting/switching, or even consciousness. But what do the authors want to prove by replicating or not their previous findings? In other words, how the manuscript may enlighten on some of the current theories of attention (e.g., inhibition, inference, attention capacity, two-stage processing)? How does this brick might contribute in a building block of knowledge? In the way that the Introduction is right now I tend to think that the main goal was to provide a tutorial on the statistical evaluation of replication studies rather than the focus on tDCS and the AB. If this was the main goal, as it seems, then I would suggest to rephrase the sentence “aside from our focus on tDCS and the AB, an important auxiliary goal of this article is to provide a tutorial on the statistical evaluation of replication studies” (p.5, line 113), because this would not be an auxiliary goal, but the main one.

We appreciate this point and have clarified in the introduction of the revised manuscript that we did not merely intend to replicate our previous findings. Replication of findings in general is important to establish credibility of scientific claims (Schmidt, 2009), and in this specific case, a potential replication of findings would be informative about the ability of tDCS to modulate the apparent bottleneck in information processing captured by the AB, and about the importance of left dorsolateral prefrontal cortex in the AB and attentional filtering more generally. This we aimed to establish. An auxiliary goal was to provide a tutorial on the statistical evaluation of replication studies. 

3) In the method, the authors wrote that “participants received anodal and cathodal tDCS in separate sessions which typically took place exactly one week apart (cf. minimum of 48 hours in London and Slagter (2021))”. Next, at some point the authors wrote “The attentional blink task was almost identical to the one used in London and Slagter (2021)”. Then, the authors wrote “The letter stream was preceded by a fixation cross (same color as the letters) presented for 1750 ms (cf. 480 ms in London and Slagter (2021)) and followed by another fixation cross (cf. none in London and Slagter (2021))”. Or the authors wrote “We used Ten20 conductive paste (Weaver and Company), because it was easier to apply concurrently with the EEG equipment (see the EEG section); London and Slagter (2021) used saline solution as a conductive medium, together with rubber straps to keep the electrodes in place.” All these sentences, and others along the text, raised some questions on what extend both procedures were, in fact, identical. Because this study is being sold as a replication study, I think that would be extremely useful to find a table in which all the main parameters as well as possible group differences between both samples are shown. A sample size of 34 or 40 participants is certainly not bad for a study using stimulation, but, due to the several experimental issues (e.g., randomization of the procedures, number of the task training trials, age, sex distribution, education level, facilities and equipment, expertise of the researcher at data collection, the fixation cross time) any minor change might really impact the outcome. For instance, in the London and Slagter (2021) the authors had 20 women out of 34 participants, as far as I understood, meaning 58%, while in the present manuscript this percentage is 72%. Therefore, I would really appreciate a complete descriptive table of both samples as well as the mean (median too, maybe) difference between both sections in the two studies and other possible confounders. The replication is not a matter of statistical analysis, only. So, I think the reader would appreciate a complete view and comparison of both research sets and participants, side-by-side. This is reinforced by the fact that neither London and Slagter (2021) and the present manuscript have a descriptive table of participants and experimental procedures such as the performance on the averaged “lag 3” in study 1 and the “lag 3” in the present study, for instance. It seems that the authors were so focused in demonstrating several methodological strategies that they underestimated very basic descriptive data on participants and procedures.

We fully agree that replication is not a matter of statistics only, as we discussed in the Discussion section, and we have tried to be as transparent as possible by listing all the differences between both studies. But we think Dr. Kluwe-Schiavon’s suggestion to structure this information in a table is an excellent idea. 

Supplementary Table S1 should list all differences between the two studies that were already discussed in the manuscript, and all of the potential differences suggested in the review, with the exception of “education level” (we did not ask this of each participant, but given that they signed up through the same university recruitment system, virtually all participants in both studies were undergraduate students), and “facilities and equipment, and […] expertise of the researcher at data collection” (both studies were ran in the same building using the same equipment, by experienced researchers trained in the same lab using the same protocol, apart from the differences already listed).

Instead of enumerating the relatively minor differences in the Task section, we now refer to this Table. However, we still mention the larger differences that could have had a bigger impact on the results in methods section, as well as the Discussion section (and now also refer to the table there as well).

As for the results, we did already include a table with descriptive statistics for the present study (Table 1), and we show the individual-participant data of attentional blink magnitude for both studies in Figure 5. However, Dr. Kluwe-Schiavon is right that we did not include any data on performance of the averaged “lag 3” condition we created in study 1, and how this compares to study 2. We now added this information to this subsection of the results.

4) The entire statistical analysis is well-written and easy to follow and Figure 6 in the results section was also great add on in the manuscript. However, in figure 6 the legend with the colors is missing. Minor issue.

We are happy to hear that this section read well, as statistical analysis sections are often not the most riveting, and this one ran rather long due to the variety of different methods we used. We are equally happy to hear that Figure 6 added additional value. We didn’t include a legend for the colors initially, as they are not essential to understand the figure (unlike Figure 5). But we now explain the colors in the caption of Figure 6.

5) Personally, I think that both the meta-analysis and the pooled analysis were good ideas. Nevertheless, some things need some additional notes. As a minor concern, the authors wrote that to deal with different sample sizes they “weighed the estimate by sample size, so the present study will have a slightly higher influence on the meta-analytic effect size”, but no reference is reported neither a clear justification to do so and the implications of it. I am not sure about this strategy either, since the difference in sample size is not that huge and by doing so the final outcome, the effect sized obtained gets a bit trick to interpret. However, another concern has to do with not using a random effect meta-analytical procedure. Although the authors have written a very interesting and important justification in this regard, along the text the authors also made clear that both experimental designs are similar, i.e., not equal. Therefore, a random effect for study could be considered or, at least, statistically proved that it is not needed (e.g., CCI). Moreover, even if the procedures would be equal, participants are certainly not, right? So, a random effect for participants could also be considered, especially because as the authors wrote “the field of electrical brain stimulation is characterized by inconsistent findings across studies” raising the questions about individual differences and characteristics, as the authors wrote “individual variability in the cortical thickness of left prefrontal cortex accounted for almost 35% of the variance in stimulation efficacy among subjects”.

We thank Dr. Kluwe-Schiavon for evaluating our meta-analysis approach with such care. Two valuable points were raised: 1) regarding weighting of the studies in the meta-analysis, and 2) whether a random-effects meta-analysis would not be more appropriate than the fixed-effects approach we take now.

Now that this has been pointed out to us, we agree that the manuscript does not clearly state the purpose of weighting the studies in the meta-analysis by sample size. We’ve now adjusted this in the paper, to make clear that the purpose is that studies that contribute more evidence affect the meta-analytic estimate more strongly. Studies with a larger sample size (or less variance, which is proportional to sample size) are able to estimate the effect more precisely, and therefore should carry more weight. This is standard practice for fixed-effects meta-analysis as far as we know, and also seems to be the default in meta-analysis software. We confirmed this is the case for the metafor package in R that we are using here, but also for the meta-analysis module in JASP, as well as the Comprehensive Meta-Analysis software. However, we did repeat the meta-analysis without weighting, which did not change the meta-analytic estimate of the correlation by a whole lot (r=-0.23 vs. -0.21), probably because the sample size of both studies does not differ that strongly.

We also agree with the reviewer that a random-effects meta-analysis could also be appropriate, depending on the analysis goal and the degree of heterogeneity that can be expected between studies. The problem here is that we have only two studies at our disposal, which is not enough to properly estimate a random-effect, nor the degree of heterogeneity. In fact, it appears that finding appropriate methods for random-effects meta-analysis of just two studies is a topic of active research in statistics (see Gonnermann et al. (2015) for example). While random-effects meta-analysis does not appear to be a viable option in our case, it might be for other researchers that have conducted more studies, which we now stress in the methods section in the adapted version of the manuscript. 

References:

Gonnermann A, Framke T, Großhennig A, Koch A. No solution yet for combining two independent studies in the presence of heterogeneity. Statistics in Medicine. 2015 Jul 20;34(16):2476.

Schmidt S. Shall we really do it again? The powerful concept of replication is neglected in the social sciences. Review of General Psychology. 2009;13: 90-100. doi: 10.1037/a0015108

---

## [Editor Report · Decision Letter 1]

3 Jan 2022

Effects of tDCS on the attentional blink revisited: A statistical evaluation of a replication attempt

PONE-D-21-17725R1

Dear Dr. Slagter,

We’re pleased to inform you that your manuscript has been judged scientifically suitable for publication and will be formally accepted for publication once it meets all outstanding technical requirements.

Kind regards,

Jorge Leite, Ph.D.

Academic Editor

PLOS ONE
---

## [Editor Report · Acceptance letter]

18 Jan 2022

PONE-D-21-17725R1 

Effects of tDCS on the attentional blink revisited: A statistical evaluation of a replication attempt 

Dear Dr. Slagter:

I'm pleased to inform you that your manuscript has been deemed suitable for publication in PLOS ONE. Congratulations! Your manuscript is now with our production department. 

Kind regards, 

on behalf of

Dr. Jorge Leite 

Academic Editor

PLOS ONE